# The implementation of a rapid sample preparation method for the detection of SARS-CoV-2 in a diagnostic laboratory in South Africa

Gert Marais [1,2]*, Michelle Naidoo [1,2], Nei-yuan Hsiao[1,2], Ziyaad Valley-Omar[1,3], Heidi Smuts[1,2], Diana Hardie[1,2]

1 Division of Medical Virology, University of Cape Town, Cape Town, Western Cape, South Africa, 2 Groote Schuur Hospital Virology Diagnostic Laboratory, National Health Laboratory Service, Cape Town, Western Cape, South Africa, 3 Groote Schuur Hospital Tissue Immunology Diagnostic Laboratory, National Health Laboratory Service, Cape Town, Western Cape, South Africa

* gert.marais16@alumni.imperial.ac.uk

**Data Availability Statement:** All relevant data are within the manuscript and its Supporting Information files.

## Abstract

The SARS-CoV-2 pandemic has resulted in shortages of both critical reagents for nucleic acid purification and highly trained staff as supply chains are strained by high demand, public health measures and frequent quarantining and isolation of staff. This created the need for alternate workflows with limited reliance on specialised reagents, equipment and staff. We present here the validation and implementation of such a workflow for preparing samples for downstream SARS-CoV-2 RT-PCR using liquid handling robots. The rapid sample preparation technique evaluated, which included sample centrifugation and heating prior to RT-PCR, showed a 97.37% (95% CI: 92.55–99.28%) positive percent agreement and 97.30% (95% CI: 90.67–99.52%) negative percent agreement compared to nucleic acid purification-based testing. This method was subsequently adopted as the primary sample preparation method in the Groote Schuur Hospital Virology Diagnostic Laboratory in Cape Town, South Africa.

## Introduction

Severe Acute Respiratory Syndrome Coronavirus 2 (SARS-CoV-2), an emergent beta-coronavirus, was identified as a novel causative agent of severe pneumonia in Wuhan, China in 2019 [1]. The capacity for person-to-person transmission was soon identified and the ensuing pandemic has caused more than seventeen million cases at the time of submission [2].

Currently, diagnostic testing for SARS-CoV-2 relies on molecular techniques, primarily reverse-transcriptase polymerase chain reaction (RT-PCR), from respiratory specimens [3]. The specialised equipment and reagents required to offer these tests at scale has placed significant strain on worldwide supply chains of reagents. Public health measures put in place in numerous countries, including travel restrictions, have further made planning for sustainable

**Funding:** The author(s) received no specific funding for this work.

**Competing interests:** The authors have declared that no competing interests exist.

service delivery difficult as laboratory stock orders may not be filled on time. These issues motivate for the use of diagnostic workflows that favour locally or readily available reagents to, at least partially, insulate supply chains from fluctuations in global demand and evolving travel limiting public health measures. To address these issues, a number of laboratories have successfully developed alternative sample preparation techniques which limit reagent needs and avoid complex nucleic acid (NA) purification protocols [4–6]. There is also a significant cost saving when the reagent-free direct heating method, as described by Fomsgaard and Rosenstierne [4], is used which will become critical if economic fallout from the pandemic intensifies. Staff shortages in the laboratory are an inevitability as social distancing requirements are implemented in concert with increasing demand for diagnostic testing. SARS-CoV-2 outbreaks in the laboratory environment may also introduce unpredictable shortages of critical staff further limiting the capacity of laboratories to offer predictable test turnaround times. The necessary influx of new staff, who may have limited training or training in a related field, can further compromise the reliability of diagnostic laboratory services as the capacity for oversight and quality control is hindered by rapidly evolving testing demands and workflow instability due to reagent shortages and potentially unreliable testing kits due to limited regulatory oversight [7]. All these factors highlight the need for automated workflows that limit the number of laboratory staff-dependent steps and in particular steps requiring specialised training. Automation further limits human error such as sample switches and cross-contamination and are generally amenable to greater degrees of workflow control due to traceable instrument log files.

A chemical reagent-free heat-based rapid sample preparation and inactivation (RSP) [8, 9] method for downstream SARS-CoV-2 RT-PCR amplification is presented here optimised for use on automated liquid handling robots.

## Materials and methods

### Ethics

Biological material of human origin was anonymised and all clinical and other personally identifiable data delinked with only study specific sample identifiers used along with sample SARS-CoV-2 assay performance data. Ethics approval for this work was granted by the University of Cape Town Human Research Ethics Committee (HREC reference number: 335/2020).

### Sample selection

Nasopharyngeal (NP) and oropharyngeal (OP) swabs sent dry or in saline to the National Health Laboratory Service Virology Diagnostic Laboratory in Groote Schuur Hospital from its standard referral area for SARS-CoV-2 testing were included. Selection of 115 samples, which tested positive, and 80 samples, which tested negative, for SARS-CoV-2 by NA purification-based commercial diagnostic assays in use at the diagnostic laboratory was done for the method validation. Spectrum bias was avoided by selecting consecutive samples that tested positive by standard testing over two discrete intervals of regular laboratory workflow. Samples that tested negative were selected randomly from the same intervals. The diagnostic assays in use were the Abbott RealTime SARS-CoV-2 Assay (Abbott Laboratories, USA) running on the Abbott m2000 RealTime system and the Allplex™ 2019-nCoV assay (Seegene, South Korea). The assays were run as per package insert. The Allplex™ 2019-nCoV assay was performed after sample NA purification using the NucliSENS® easyMag® (bioMérieux, France) as per package insert.

## Rapid sample preparation

Standard diagnostic testing sample preparation included placing NP or OP swabs in a 2ml Sarstedt sample tube containing 1.5ml autoclaved 0.9% saline. If both a NP and OP swab or multiple swabs of the same type was received, they were combined in a single tube. The swabs were cut to fit in the tube. The tube was then vortexed for 10 seconds. The saline was used as the sample input for downstream assays after which the tube was stored at 4˚C. Stored tubes from diagnostic samples were available for inclusion in the study.

Selected sample tubes were centrifuged at 16 000 *g* for 5 minutes and 50μl of the supernatant was then pipetted into the wells of a 96-well PCR plate. The PCR wells were capped and the plate incubated on a thermocycler at 98˚C for 5 minutes followed by 4˚C for 2 minutes. The PCR plate was then briefly centrifuged and placed on a dedicated QIAgility (Qiagen, Germany) liquid handling instrument for sample-addition.

## RT-PCR after rapid sample preparation

Concurrent with sample preparation, a second dedicated QIAgility instrument was used for Allplex™ 2019-nCoV assay master mix preparation and aliquoting into appropriate 8-well PCR strips (Bio-Rad Laboratories, USA). Following master mix preparation, the PCR strips were transferred to the sample-addition QIAgility instrument. The sample input volume and master mix constituents are shown in Table 1.

After sample addition, the PCR strips were sealed and briefly centrifuged before being loaded on a CFX96™ Real-Time PCR Detection System (Bio-Rad Laboratories, USA). The real-time PCR cycling parameters recommended by the Allplex™ 2019-nCoV assay package insert were used unchanged. Real-time data analysis was performed using the 2019-nCoV Viewer for Real time Instruments V3 (Ver 3.18.005.003) software as per the Allplex™ 2019-nCoV assay package insert.

If the internal control (RP-IC) was not detected with a cycle threshold (Ct) value <40 and no SARS-CoV-2 targets were detected, the test was deemed invalid and the primary sample was retested with a decreased sample volume input, 2μl instead of 3μl. The remainder of the protocol was unchanged.

## Repeatability and analytical sensitivity

Inter-assay reproducibility was assessed using 8 samples with Envelope (E) gene Ct values ranging between 17.16 and 35.63, which were tested in triplicate 7 days after initial testing. Intra-assay reproducibility was assessed by repeating 16 samples in triplicate. Samples were stored at 4˚C while awaiting repeat testing. To assess relative analytical sensitivity, one sample was selected and serially diluted with saline and tested with multiple replicates at dilutions specifically selected to allow calculation of the analytical sensitivity of the Allplex™ 2019-nCoV

**Table 1. RT-PCR reaction preparation.**

|  | Volume per reaction (μl) |
|---|---|
| RNase-free Water | 11.1 |
| 2019-nCoV MOM (primer and probe mix) | 6 |
| 5X Real-time One-step Buffer | 6 |
| Real-time One-step Enzyme | 2.4 |
| Internal control (RP-IC) | 1.5 |
| Sample after centrifugation and heating | 3 |
| Total volume | 30 |

assay after NA purification and RSP. The dilution at which SARS-CoV-2 RNA could be detected with 95% confidence was determined for each method by Probit analysis. The absolute analytical sensitivity of the RSP method was then calculated based on the relative analytical sensitivity compared to NA purification-based detection. The absolute analytical sensitivity for NA purification-based detection is reported in the Allplex™ 2019-nCoV assay package insert.

### Statistical analysis and graphics

Data visualisation and statistical analysis, including paired t-tests for comparison of target Ct values, a Fisher's exact test for statistical significance determination of the positive percent agreement (PPA) and negative percent agreement (NPA) with NA extraction-based testing and the Wilson/Brown method for 95% confidence interval determination, was done using GraphPad Prism version 8.4.2 for macOS, GraphPad Software, San Diego, California USA, www.graphpad.com.

## Results and discussion

The RSP method validation included 115 serially collected samples which tested positive and 80 randomly selected samples from the same period which tested negative for SARS-CoV-2 by NA purification-based testing. After testing with the RSP method, repeat testing with a decreased sample volume was required for 20 of the 195 (10.26%) samples due to detection of neither SARS-CoV-2 targets nor the internal control. One sample could not be tested using the RSP method due to excessive viscosity from nasopharyngeal swab breakdown. Repeat testing failed to generate a result for 6 samples possibly due to sample-specific PCR inhibition. The Allplex™ 2019-nCoV assay result after RSP correlated with that of NA purification-based testing for 111 positive and 72 negative samples as shown in Table 2. No result could be generated for 7 of 195 (3.59%) samples. Raw data is shown in the S1 Appendix.

The PPA and NPA of the RSP method with NA purification-based testing for SARS-CoV-2 demonstrated a P value of <0.0001. The PPA of the RSP method was 97.37% (95% CI: 92.55–99.28%) and the NPA 97.30% (95% CI: 90.67–99.52%). The 7 samples, for which no result could be generated by RSP due to repeated invalid results or sample unsuitability, were excluded from this analysis as standard laboratory practice designates samples for NA purification-based testing in cases of RSP failure.

The Ct values of individual targets of the Allplex™ 2019-nCoV assay were assessed for samples prepared by NucliSENS® easyMag® NA purification and RSP. The E gene, RNA-dependent RNA-polymerase (RdRp) gene and Nucleocapsid (N) gene targets had Ct values that were significantly different with a P value of <0.0001 (Fig 1). The mean difference in Ct values

**Table 2. Contingency table used for positive and negative percent agreement with NA purification-based testing calculation.**

| | Positive SARS-CoV-2 | Negative SARS-CoV-2 |
|---|---|---|
| | Abbott RealTime SARS-CoV-2 Assay or Seegene Allplex™ 2019-nCoV Assay | Abbott RealTime SARS-CoV-2 Assay or Seegene Allplex™ 2019-nCoV Assay |
| | NA Purification | NA Purification |
| Positive SARS-CoV-2, RSP method, Seegene Allplex™ 2019-nCoV Assay | 111 | 2 |
| Negative SARS-CoV-2, RSP method, Seegene Allplex™ 2019-nCoV Assay | 3 | 72 |

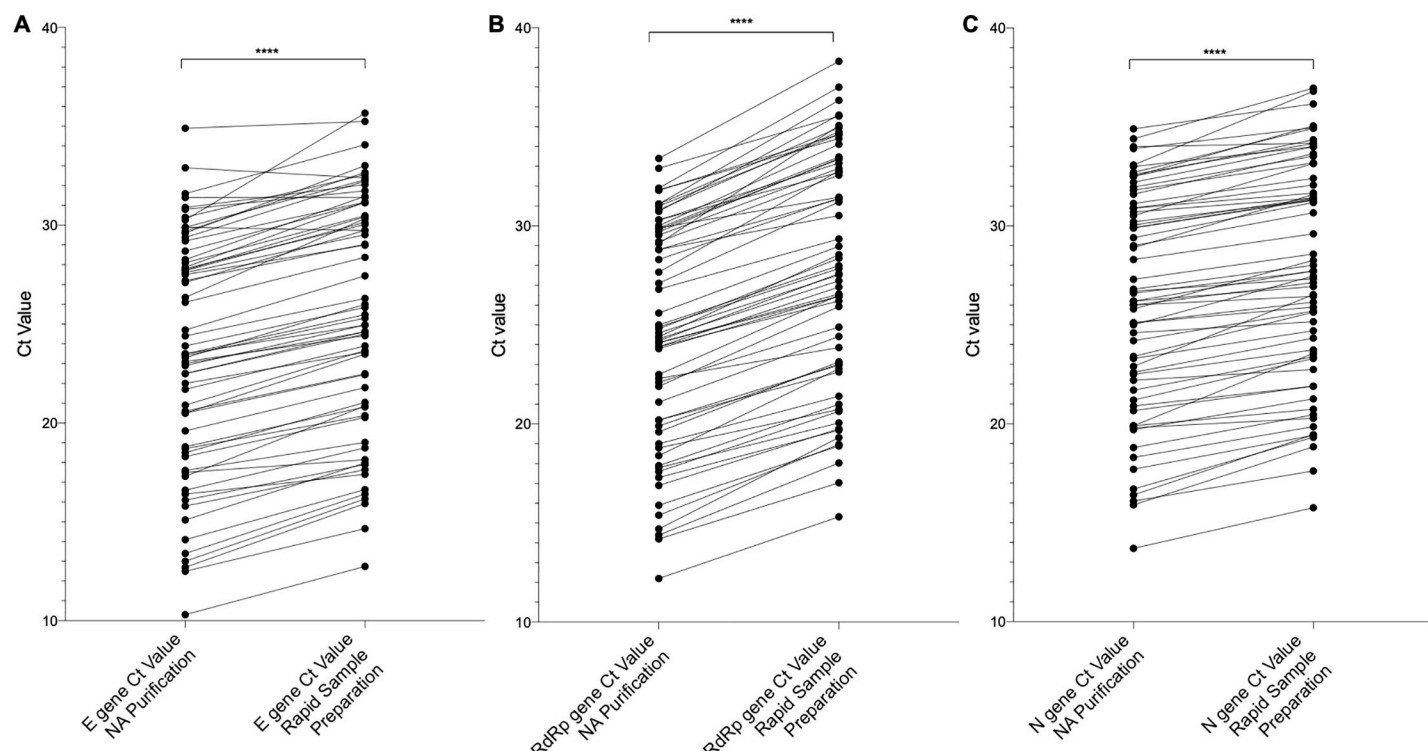

**Fig 1. Comparison of target Ct values after RSP and NucliSENS® easyMag NA purification.** The Ct values for the SARS-CoV-2 (A) Envelope (E), (B) RNA-dependent RNA-polymerase (RdRp) and (C) Nucleocapsid (N) gene targets are shown for samples tested with the Allplex™ 2019-nCoV assay after NucliSENS® easyMag® NA purification and RSP. The difference in generated Ct values was found to be statistically significant in each case with a P value of <0.0001 as determined by paired t-test.

between RSP and NA purification was 2.148 (95% CI: 1.909–2.387) for the E gene, 3.271 (95% CI: 3.037–3.506) for the RdRp gene and 1.608 (95% CI: 1.407–1.809) for the N gene, with RSP demonstrating a higher mean Ct value in each case.

The relative performance of the Abbott RealTime SARS-CoV-2 assay and the Allplex™ 2019-nCoV assay after RSP is shown in Fig 2. The Abbott assay reports cycle number (CN) values which are not equivalent to Ct values and thus are not directly comparable.

The single false negative result from the RSP method when compared to NucliSENS® easyMag® NA purification was from a sample that only tested positive for one of the three Allplex™ 2019-nCoV targets, the N gene, with a Ct value of 36.7. The two false negatives from the RSP method when compared to the Abbott RealTime SARS-CoV-2 Assay, which includes NA purification, had high CN values. However, samples with higher CN values were detected thus sample-specific inhibition may also have played a role.

There were two false positive results from the RSP method when compared to the Abbott RealTime SARS-CoV-2 Assay. A single target was detected in both cases with Ct values above 35. This may represent contamination events or the samples may have viral RNA at levels near the limit of detection for both assays. NA contamination in the laboratory is monitored for by frequent testing of environmental swabs and reagent blanks. Multiple negative controls are also included in each run.

The intra-assay repeatability assessment of mean Ct values for the three Allplex™ 2019-nCoV targets showed a coefficient of variance of 1.14%. The inter-assay repeatability assessment of mean Ct values after 7 days of sample storage showed a coefficient of variance of 1.27%.

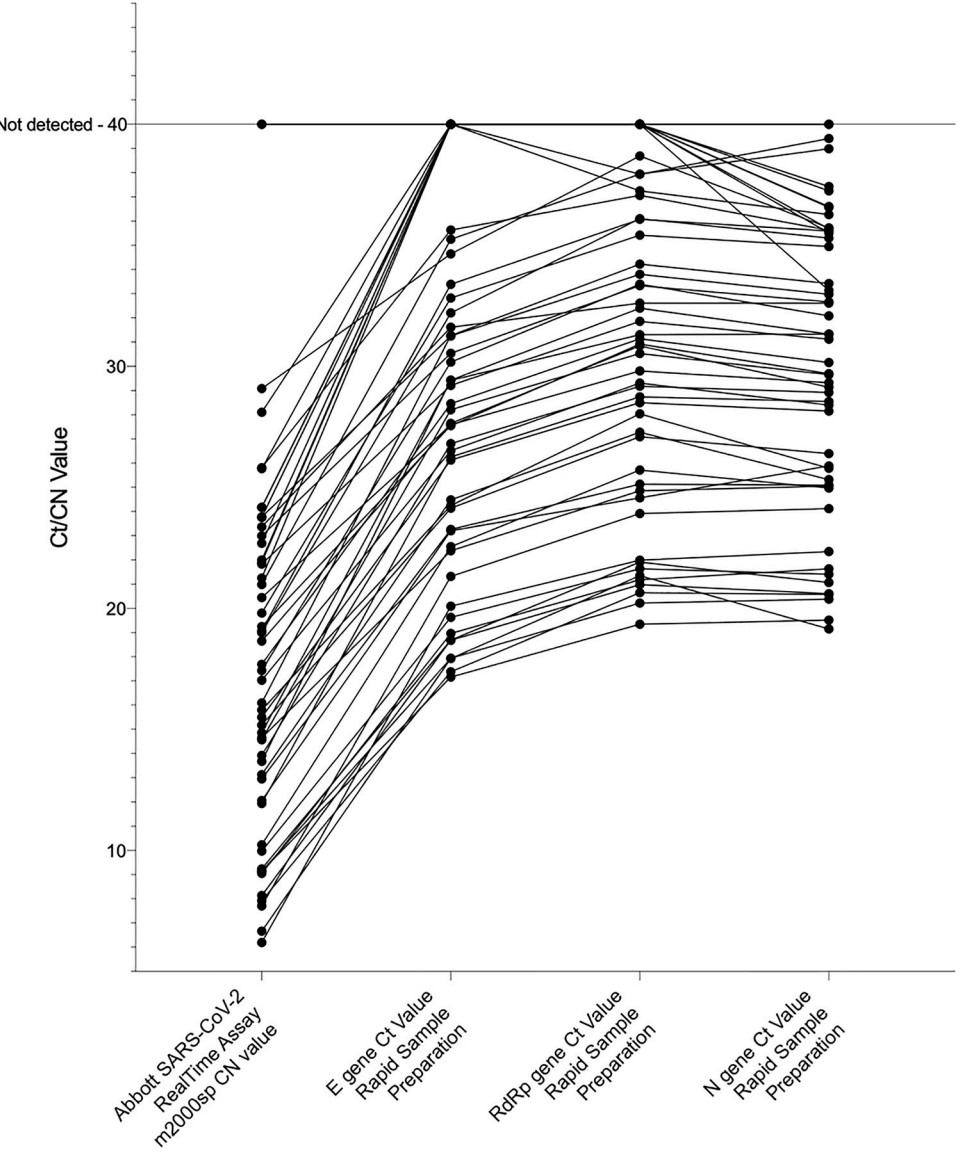

**Fig 2. Comparison of target Ct and CN values after RSP and testing with the Abbott RealTime SARS-CoV-2 assay.** The Ct values for the SARS-CoV-2 Envelope (E), RNA-dependent RNA-polymerase (RdRp) and Nucleocapsid (N) gene targets are shown for samples tested with the Allplex™ 2019-nCoV assay after RSP and CN values after testing with the Abbott RealTime SARS-CoV-2 assay. A plotted CN or Ct value of 40 indicates that detectable amplification did not occur. The Abbott assay CN values are assay specific and not directly comparable to Ct values, but are shown to demonstrate the performance of the spectrum of selected samples.

The relative analytical sensitivity of the Allplex™ 2019-nCoV assay after RSP was found to be 807 RNA copies per reaction. This was calculated from the 8.07-fold decrease in analytical sensitivity of the RSP method compared to NucliSENS® easyMag® NA purification-based testing, which has an analytical sensitivity of 100 RNA copies per reaction as per the Allplex™ 2019-nCoV assay package insert. The relative decrease was determined by serially diluting and testing a sample with multiple replicates as shown in Table 3. This relative loss in analytical sensitivity can largely be explained by the smaller sample input volume for RSP. NucliSENS® easyMag® NA purification concentrates sample nucleic acids by a factor of approximately 2,

**Table 3. Relative analytical sensitivity assessment.**

| Dilution | Replicates | Seegene Allplex™ 2019-nCoV Assay RSP Method Percentage of Samples Positive | Seegene Allplex™ 2019-nCoV Assay NA Purification Percentage of Samples Positive |
|---|---|---|---|
| 1:20 | 24 | 100% | Not done |
| 1:40 | 24 | 95.8% | Not done |
| 1:80 | 24 | 70.8% | Not done |
| 1:120 | 24 | 58.3% | Not done |
| 1:160 | 24 | 41.7% | Not done |
| 1:200 | 10 | Not done | 100% |
| 1:320 | 24 | 33.3% | Not done |
| 1:400 | 10 | Not done | 100% |
| 1:500 | 10 | Not done | 90% |
| 1:625 | 10 | Not done | 70% |
| 1:2000 | 10 | Not done | 60% |
| 1:5000 | 10 | Not done | 30% |

based on sample input versus elution volume. Additionally, the Allplex™ 2019-nCoV assay input volume after NA purification is 8μl versus the 3μl sample input volume for RSP. Thus, the expected loss in analytical sensitivity would be 5.3-fold which is comparable to the experimentally determined loss of 8.07-fold and suggests that sample inhibition plays a minor role. Raw data is shown in the S2 Appendix.

The performance characteristics were deemed acceptable for clinical diagnostic use in the Groote Schuur Hospital Virology Diagnostic Laboratory and allowed the laboratory to increase the number of samples tested daily by a factor of 5–10 due to the decreased supply chain dependence and simplified workflow. While large quantities of some consumables were still required, such as liquid handling robot tips for the QIAgility instruments, the availability of generic alternatives and the fact that they are neither SARS-CoV-2 specific nor universally required made consumable depletion less of a concern. The reduced processing time further facilitated a more rapid test turnaround time which was beneficial for in-hospital infection control. A stable workflow, not subject to reagent availability dependent variations, also decreased laboratory errors and may allow for improved clinical planning as a result of a stable test turnaround time.

Prior to the automation described in this protocol, earlier versions of the RSP method were susceptible to fluctuating failure rates. This was largely due to human errors arising from staff shortages and rising test volumes. A simple automated workflow was needed to enable staff with minimal molecular experience to be able to perform testing reliably. In particular the time intervals between assay steps and how thoroughly the master mix was mixed prior to aliquoting were identified as sources of assay performance variation. This operator dependency and fluctuating staff availability motivated for the further automation of the process with liquid handling robots and ultimately the validation described here.

The laboratory approach to result interpretation was also affected by the implementation of the RSP method. The approach to NucliSENS® easyMag® NA purification-prepared samples involved release of numerous inconclusive results, despite multiple target amplification at times, due to the known capacity for sample contamination both on the easyMag® instrument and during processing of swabs. The known decrease in sensitivity of the RSP method and the lack of use of the easyMAG® open system for processing, decreased the number of inconclusive results released by our laboratory.

NA purification is the gold-standard in sample processing for RT-PCR, however, in the setting of a pandemic with significant pressures on reagent supply chains and the need for a rapid increase in testing capacity, the RSP method described here presented a reasonable alternative and has been implemented as the primary sample preparation method in the Groote Schuur Hospital Virology Diagnostic Laboratory in South Africa.

## Supporting information

**S1 Appendix. Sample cycle threshold and cycle number values for SARS-CoV-2 targets and internal controls.** The cycle threshold (Ct) and cycle number (CN) values of assay targets and internal controls from the Allplex™ 2019-nCoV and Abbott RealTime SARS-CoV-2 assays respectively are shown for samples used. The mastermix protocol used is also shown. RSP: Rapid sample preparation and inactivation.
(XLSX)

**S2 Appendix. Sample cycle threshold values at dilutions used for analytical sensitivity determination.** The cycle threshold (Ct) values for the Allplex™ 2019-nCoV assay targets and internal control at dilutions used in the determination of the analytical sensitivity of the rapid sample preparation and inactivation (RSP) method relative to nucleic acid purification.
(XLSX)

## Author Contributions

**Conceptualization:** Gert Marais, Michelle Naidoo, Heidi Smuts, Diana Hardie.

**Data curation:** Gert Marais.

**Formal analysis:** Gert Marais, Nei-yuan Hsiao.

**Investigation:** Gert Marais, Michelle Naidoo, Ziyaad Valley-Omar, Diana Hardie.

**Methodology:** Gert Marais, Michelle Naidoo, Nei-yuan Hsiao, Ziyaad Valley-Omar, Heidi Smuts, Diana Hardie.

**Project administration:** Gert Marais.

**Software:** Gert Marais.

**Supervision:** Diana Hardie.

**Validation:** Gert Marais, Nei-yuan Hsiao.

**Visualization:** Gert Marais.

**Writing – original draft:** Gert Marais.

**Writing – review & editing:** Michelle Naidoo, Nei-yuan Hsiao, Ziyaad Valley-Omar, Heidi Smuts, Diana Hardie.

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
