## [Decision Letter · Decision Letter 0]

1 Sep 2020

PONE-D-20-24160

The implementation of a rapid sample preparation method for the detection of SARS-CoV-2 in a diagnostic laboratory in South Africa

PLOS ONE

Dear Dr. Marais,

Thank you for submitting your manuscript to PLOS ONE. After careful consideration, we feel that it has merit but does not fully meet PLOS ONE’s publication criteria as it currently stands. Therefore, we invite you to submit a revised version of the manuscript that addresses the points raised during the review process.

This work is interesting and relevant for the worldwide coronavirus testing.

In addition to the comments of both reviewers, with which I agree, there are a few (minor) comments that I would like to add, to further improve the quality of your manuscript.

We look forward to receiving your revised manuscript.

Kind regards,

Sylvia Maria Bruisten, Ph.D

Academic Editor

PLOS ONE

Journal Requirements:

Additional Editor Comments (if provided):

1. Table 4 and the paragraph where these data are described (page 12) are not completely clear to me. Samples were serially diluted and tested in several replicates (for example 10 or 24). Testing was however for dilutions 1:20 to 1:160 and 1:320 only performed with the RSP method whereas for all other dilutions it was performed with the NA purification method. This does not allow a direct comparison of the sensitivities of the RSP and NA methods. Why were the dilutions not tested in both ways, for example 12 replicates for each dilution for both RSP and NA?

2. Please avoid starting a sentence with a number (for example in lines 161 and 200). Please rephrase these sentences.

3. Line 212: please remove 'are shown' at the end of the sentence.

Reviewers' comments:

Reviewer's Responses to Questions

**Comments to the Author**

1. Is the manuscript technically sound, and do the data support the conclusions?

Reviewer #1: Yes

Reviewer #2: Yes

2. Has the statistical analysis been performed appropriately and rigorously? 

Reviewer #1: Yes

Reviewer #2: N/A

3. Have the authors made all data underlying the findings in their manuscript fully available?

Reviewer #1: No

Reviewer #2: Yes

4. Is the manuscript presented in an intelligible fashion and written in standard English?

Reviewer #1: Yes

Reviewer #2: Yes

5. Review Comments to the Author

Reviewer #1: This manuscripts presents data evaluating procedures to omit the need of nucleic acaid extraction from clinical NP/OP swab samples prior to performing molecular testing for Sars-CoV-2 detection. All results for extraction free procedures are compared to established extraction method (used as gold standards).

The results demonstrate that the extraction free procedure leads to some loss of analytical sensitivity, in particular for samples harbouring a low viral load (high Ct values). In general Ct values for samples without extraction are higher as compared to extracted samples. This could either be due to a reduced amplification efficiency (or even inhibition) or a smaller equivalent of the clinical sample used as input into the PCR reaction.

Specific questions:

1. It would be relevant to present the Ct values of the internal control of all samples w/wo extraction listed in appendix 1 and 2, as this will give insight in the effect of (leaving out) extraction on PCR efficiency / inhibition.

2. 6 previously negative samples were left out from the analysis because the IC failed (even after repeat testing upon dilution). These samples should not have been left out from the analysis but included in table 3, because the information is very relevant in judging the appropriateness and feasbility of the extraction free protocol : The results demonstrate that PCR inhibition was present in 6/185 samples (3%).

3. Nucleic acid extraction using chaotropic agents (Guanidinium salts) result in virus inactivation (loss of infectivity). The extraction-free protocol is based on a 5 minute incubation at 98C. Did the investigators perform any experiments to study the effect of this temperature treatment on sample infectivity (bio-safety). Samples which are manipulated on a QIAgility liquid handling system, given the ‘open environment’ of such a system that lacks HEPA filtering of exhausted air, should be proven to be non-infectious

4. The authors indicate that automation of the PCR setup process significantly reduced robustness of assay performance by reducing the frequency of invalid results. This is just mentioned in the discussion without supporting data. What is menat by invalid results (PC negative / NC positive / IC negative???) and how are these data used in the manuscript (in particular in the S1 appendix)?

5. In the methods section it is described that PCR setup was don using an liquid handling system whereas in the discussion it is mentioned that manual setup was don for at least part of the experiments (and that this is caused operator dependency in the quality of the results). How did these differences in PCR setup procedures affect the overall results and conclusion on the comparison of extraction free procedures to the gold standard methods?

Reviewer #2: This manuscript by Marais and co-workers describes a rapid automated sample preparation method for the detection of SARS-CoV-2. This information is important as limited availability of general nucleic acid purification reagents have impacted SARS-CoV-2 testing worldwide.

There are a number of issues that need to be addressed:

1) The authors mention (lines 131-134) that if the internal control failed (ct <40) the sample was repeated with less sample input. They mention (lines 172-174) in 6 negative samples this was the case after repeat testing. They do not mention however the percentage of samples overall that failed internal control (ct<40) in the initial analysis. This is important because if this percentage is high it would mean a significant increased workload for retesting.

2) The limited availability of reagents was the main reason for this study. The authors may want to comment on availability of consumables for the QIAgility systems.

3) The authors estimate PPA (lines 193-202) based on the mean difference in Ct values between the Nuclisens and RSP method and adding these numbers to Ct values from previously determined samples. They argue that if this newly calculated Ct value was above 40 the sample would be negative if they had used the RSP method. By doing this the authors assume that the relation between the amount of RNA and the Ct value is linear over the entire range of RNA concentrations. The authors do not show this linear correlation. Especially at high Ct values this correlation is almost never linear and generally very variable. In my opinion this method cannot be used to determine the PPA of the RSP method and the authors should delete this part from the manuscript

4) Since the values from the Abbott M2000 system cannot be compared to the Ct values from the Seegene PCR due to intrinsic different analysis method I fail to see what information is added by figure 2.

5) The authors mention that the loss of analytical sensitivity of at least 8 fold was acceptable for clinical application. It is unclear however which criteria played a role in this consideration.

6) Furthermore they mention that the Seegene assay has an analytical sensitivity of 100 RNA copies/reaction with the nuclisense method (and thus > 800 c/reaction for the RSP method). This analytical sensitivity seems rather low compared to other molecular assays which are in the range of 1-50 (see below refs). This should also be taken into consideration with remark 5)

Corman VM, Landt O, Kaiser M, Molenkamp R, Meijer A, Chu DK, Bleicker T,

Brünink S, Schneider J, Schmidt ML, Mulders DG, Haagmans BL, van der Veer B, van den Brink S, Wijsman L, Goderski G, Romette JL, Ellis J, Zambon M, Peiris M, Goossens H, Reusken C, Koopmans MP, Drosten C. Detection of 2019 novel

coronavirus (2019-nCoV) by real-time RT-PCR. Euro Surveill. 2020, Jan;25(3):2000045

van Kasteren PB, van der Veer B, van den Brink S, Wijsman L, de Jonge J, van

den Brandt A, Molenkamp R, Reusken CBEM, Meijer A. Comparison of seven

commercial RT-PCR diagnostic kits for COVID-19. J Clin Virol. 2020

Jul;128:104412. doi: 10.1016/j.jcv.2020.104412

Iglói Z, Leven M, Abdel-Karem Abou-Nouar Z, Weller B, Matheeussen V, Coppens

J, Koopmans M, Molenkamp R. Comparison of commercial realtime reverse

transcription PCR assays for the detection of SARS-CoV-2. J Clin Virol. 2020

Aug;129:104510. doi: 10.1016/j.jcv.2020.104510

6. PLOS authors have the option to publish the peer review history of their article (what does this mean?). If published, this will include your full peer review and any attached files.

Reviewer #1: No

Reviewer #2: **Yes: **Richard Molenkamp

---

## [Author Response · Author response to Decision Letter 0]

28 Sep 2020

Editor’s comments

Comment:

1. Table 4 and the paragraph where these data are described (page 12) are not completely clear to me. Samples were serially diluted and tested in several replicates (for example 10 or 24). Testing was however for dilutions 1:20 to 1:160 and 1:320 only performed with the RSP method whereas for all other dilutions it was performed with the NA purification method. This does not allow a direct comparison of the sensitivities of the RSP and NA methods. Why were the dilutions not tested in both ways, for example 12 replicates for each dilution for both RSP and NA?

Response:

The table shows the same sample, thus allowing direct comparison, that was serially diluted in the range 1:20 to 1:5000. Due to the expected greater sensitivity of NA purification, it was deemed unnecessary to perform multiple replicates at a dilution of less than 1:200 as all replicates tested at 1:200 and 1:400 were detected. With the RSP method, performing additional replicates at a dilution of greater than 1:320, where 33% of replicates were detected, was deemed unnecessary as the goal was to determine the dilution at which targets would be detected with 95% confidence. 

The table thus shows the data that was required to determine the dilution at which a specific sample could be detected with 95% confidence using the RSP method and NA purification. This value could then be compared. 

The methods section of the manuscript was revised to clarify the selection of sample dilutions. 

Comment:

2. Please avoid starting a sentence with a number (for example in lines 161 and 200). Please rephrase these sentences.

Response:

The manuscript was appropriately revised. 

Comment:

3. Line 212: please remove 'are shown' at the end of the sentence.

Response:

The manuscript was appropriately revised. 

Reviewer 1 Comments 

Comment:

1. It would be relevant to present the Ct values of the internal control of all samples w/wo extraction listed in appendix 1 and 2, as this will give insight in the effect of (leaving out) extraction on PCR efficiency / inhibition.

Response:

The tables presented in the appendixes were updated with the internal control values for each sample tested to provide insight into PCR inhibition and extraction efficiency. 

Comment:

2. 6 previously negative samples were left out from the analysis because the IC failed (even after repeat testing upon dilution). These samples should not have been left out from the analysis but included in table 3, because the information is very relevant in judging the appropriateness and feasibility of the extraction free protocol : The results demonstrate that PCR inhibition was present in 6/185 samples (3%).

Response:

The samples which failed testing by the RSP method or could not be tested (3.59%) were excluded from table 3 as the standard testing procedure would designate these samples for retesting by an alternative method. Thus assigning these samples as either false negatives or false positives would be inappropriate as these would not be the results reported by the laboratory. However, the manuscript was revised to more clearly highlight this failure rate.

In terms of a feasibility assessment, we feel the current PPA and NPA values along with a reported failure rate is a more reasonable way of presenting the data than reduction of all data to the PPA and NPA. 

Comment:

3. Nucleic acid extraction using chaotropic agents (Guanidinium salts) result in virus inactivation (loss of infectivity). The extraction-free protocol is based on a 5 minute incubation at 98C. Did the investigators perform any experiments to study the effect of this temperature treatment on sample infectivity (bio-safety). Samples which are manipulated on a QIAgility liquid handling system, given the ‘open environment’ of such a system that lacks HEPA filtering of exhausted air, should be proven to be non-infectious

Response:

The sample infectivity was deemed to be ablated after heat treatment at 98 degrees C for 5 minutes based on available publications. Batéjat et al. (2020) demonstrated inactivation of SARS-CoV-2 after heat treatment at 95�C for 3 minutes. Further, Saknimit et al. (1988) demonstrated heat inactivation of coronaviruses other than SARS-CoV-2 beyond specific quantification after heat treatment at 80�C for 1 minute. This literature is referenced in the revised manuscript. 

References:

Batéjat, C., Grassin, Q. and Manuguerra, J.C., 2020. Heat inactivation of the Severe Acute Respiratory Syndrome Coronavirus 2. bioRxiv.

Saknimit, M., Inatsuki, I., Sugiyama, Y. and Yagami, K.I., 1988. Virucidal efficacy of physico-chemical treatments against coronaviruses and parvoviruses of laboratory animals. Experimental animals, 37(3), pp.341-345.

Comment:

4. The authors indicate that automation of the PCR setup process significantly reduced robustness of assay performance by reducing the frequency of invalid results. This is just mentioned in the discussion without supporting data. What is meant by invalid results (PC negative / NC positive / IC negative???) and how are these data used in the manuscript (in particular in the S1 appendix)?

Response:

Invalid results in this context specifically refers to samples that lack both internal control amplification and SARS-CoV-2 target amplification. This definition was more clearly presented in the methods section of the revised manuscript. 

Prior to implementation of the automated method, staff shortages were frequent due to rapidly scaling testing demand and intermittent quarantining of staff. Thus staff with minimal molecular experience needed to be trained and staff frequently returned after extended absences. We noticed that these events frequently correlated with an increase in invalid rate but a formal critical assessment of the early pandemic SARS-CoV-2 testing performance of our laboratory is beyond the intended purpose of this work. The anecdotal data of fluctuating invalid rate and operator dependency as a potential aetiology motivated for the initiation of this research. 

The manuscript and appendixes were revised to include only data directly involved in the generation of the discussed results. The paragraph discussing the motivation for assay automation was revised to remove specific references to previous assay results and protocols and presented as a general discussion of the events leading to the research.

Comment:

5. In the methods section it is described that PCR setup was don using an liquid handling system whereas in the discussion it is mentioned that manual setup was don for at least part of the experiments (and that this is caused operator dependency in the quality of the results). How did these differences in PCR setup procedures affect the overall results and conclusion on the comparison of extraction free procedures to the gold standard methods?

Response:

No results from the manual set-up of the RSP method, which only occurred for prior version of the method used before the initiation of this research, were included. All data from versions of the RSP method not used in the direct generation of the presented results were removed from the appendixes in the updated manuscript. This was initially included to provide insight into the progression of method development. 

Reviewer 2 Comments 

Comment:

1) The authors mention (lines 131-134) that if the internal control failed (ct <40) the sample was repeated with less sample input. They mention (lines 172-174) in 6 negative samples this was the case after repeat testing. They do not mention however the percentage of samples overall that failed internal control (ct<40) in the initial analysis. This is important because if this percentage is high it would mean a significant increased workload for retesting.

Response:

The manuscript was revised to more clearly show the assay failure rate and steps taken to produce results when the primary protocol failed to produce a result.

Comment:

2) The limited availability of reagents was the main reason for this study. The authors may want to comment on availability of consumables for the QIAgility systems.

Response:

The availability of QIAgility consumables is discussed in the revised manuscript. 

Comment:

3) The authors estimate PPA (lines 193-202) based on the mean difference in Ct values between the Nuclisens and RSP method and adding these numbers to Ct values from previously determined samples. They argue that if this newly calculated Ct value was above 40 the sample would be negative if they had used the RSP method. By doing this the authors assume that the relation between the amount of RNA and the Ct value is linear over the entire range of RNA concentrations. The authors do not show this linear correlation. Especially at high Ct values this correlation is almost never linear and generally very variable. In my opinion this method cannot be used to determine the PPA of the RSP method and the authors should delete this part from the manuscript.

Response:

This part of the manuscript was excluded, as suggested, from the revised manuscript. 

Comment:

4) Since the values from the Abbott M2000 system cannot be compared to the Ct values from the Seegene PCR due to intrinsic different analysis method I fail to see what information is added by figure 2.

Response:

While the Abbott RealTime SARS-CoV-2 reported CN values are not directly comparable, they are still based on a real-time PCR cycle threshold value and thus we feel the distribution of values is relevant to the data. If only samples with low CN values were used in the validation, for example, the PPA would likely be greater than that reported. 

Additionally, while it would be inappropriate to perform any more in-depth analysis due to the disparate test specifics, for operators of the Abbott RealTime SARS-C0V-2 assay we believe a general impression of relative performance as presented by Figure 2 may be valuable. 

Comment:

5) The authors mention that the loss of analytical sensitivity of at least 8 fold was acceptable for clinical application. It is unclear however which criteria played a role in this consideration.

Response:

The primary determinant of acceptability for clinical application of the assay was the PPA and NPA. The analytical sensitivity calculated here allows assessment of the relative contribution of PCR inhibition and sample input volume as the aetiology of differing performance but was not used as the determinant of assay acceptability.

Comment:

6) Furthermore they mention that the Seegene assay has an analytical sensitivity of 100 RNA copies/reaction with the nuclisense method (and thus > 800 c/reaction for the RSP method). This analytical sensitivity seems rather low compared to other molecular assays which are in the range of 1-50 (see below refs). This should also be taken into consideration with remark 5)

Response:

While the Seegene reported analytical sensitivity may be poorer than that of other molecular assays, the PPA and NPA were determined from comparison to both the Seegene and Abbott assays. Further, we did not notice a marked difference in performance of the RSP method compared to NA purification relative to its performance compared to the Abbott system as presented in the appendixes. Additionally, the poorer limit of detection still falls below the reported critical value of 6.63 log10 RNA copies/ml associated with infectivity proposed by van Kampen et al. (2020).

Reference: 

van Kampen, J.J., van de Vijver, D.A., Fraaij, P.L., Haagmans, B.L., Lamers, M.M., Okba, N., van den Akker, J.P., Endeman, H., Gommers, D.A., Cornelissen, J.J. and Hoek, R.A., 2020. Shedding of infectious virus in hospitalized patients with coronavirus disease-2019 (COVID-19): duration and key determinants. medRxiv.

---

## [Decision Letter · Decision Letter 1]

7 Oct 2020

PONE-D-20-24160R1

The implementation of a rapid sample preparation method for the detection of SARS-CoV-2 in a diagnostic laboratory in South Africa

PLOS ONE

Dear Dr. Marais,

Thank you for submitting your manuscript to PLOS ONE. After careful consideration, we feel that it has merit but does not fully meet PLOS ONE’s publication criteria as it currently stands. Therefore, we invite you to submit a revised version of the manuscript that addresses the points raised during the review process.

There are two minor points that will further improve the manuscript. (see below).

We look forward to receiving your revised manuscript.

Kind regards,

Sylvia Maria Bruisten, Ph.D

Academic Editor

PLOS ONE

Additional Editor Comments (if provided):

The revised version shows good improvements in manuscript and supplementary files. Most points were answered to satisfaction.

There are two (minor) points that can still improve the manuscript:

1. Table 2 is redundant since here exactly the same mixture scheme is used as in Table 1, with the difference that only 2 μl input in stead of 3 μl was used (which is compensated for by the water volume). I therefor advise to remove Table 2 and to add in the text after 'with a decreased sample volume' '2 μl in stead of 3 μl' (page 7, line 134).

2. Please replace 'greater' by 'higher' before 'mean Ct value'

Reviewers' comments:

Reviewer's Responses to Questions

**Comments to the Author**

1. If the authors have adequately addressed your comments raised in a previous round of review and you feel that this manuscript is now acceptable for publication, you may indicate that here to bypass the “Comments to the Author” section, enter your conflict of interest statement in the “Confidential to Editor” section, and submit your "Accept" recommendation.

Reviewer #2: All comments have been addressed

2. Is the manuscript technically sound, and do the data support the conclusions?

Reviewer #2: Yes

3. Has the statistical analysis been performed appropriately and rigorously? 

Reviewer #2: Yes

4. Have the authors made all data underlying the findings in their manuscript fully available?

Reviewer #2: Yes

5. Is the manuscript presented in an intelligible fashion and written in standard English?

Reviewer #2: Yes

6. Review Comments to the Author

Reviewer #2: (No Response)

7. PLOS authors have the option to publish the peer review history of their article (what does this mean?). If published, this will include your full peer review and any attached files.

Reviewer #2: No

---

## [Author Response · Author response to Decision Letter 1]

7 Oct 2020

Thank you for the review of our manuscript. We have prepared responses to the comments provided in addition to a revised manuscript. 

Editor’s comments

Comment:

1.Table 2 is redundant since here exactly the same mixture scheme is used as in Table 1, with the difference that only 2 μl input in stead of 3 μl was used (which is compensated for by the water volume). I therefor advise to remove Table 2 and to add in the text after 'with a decreased sample volume' '2 μl in stead of 3 μl' (page 7, line 134).

Response:

The manuscript has been appropriately updated.

Comment:

2. Please replace 'greater' by 'higher' before 'mean Ct value'

Response:

The manuscript has been appropriately updated.

---

## [Editor Report · Decision Letter 2]

8 Oct 2020

The implementation of a rapid sample preparation method for the detection of SARS-CoV-2 in a diagnostic laboratory in South Africa

PONE-D-20-24160R2

Dear Dr. Marais,

We’re pleased to inform you that your manuscript has been judged scientifically suitable for publication and will be formally accepted for publication once it meets all outstanding technical requirements.

This includes to re-number the Table, after the deletion of Table 2.

Kind regards,

Sylvia Maria Bruisten, Ph.D

Academic Editor

PLOS ONE

Additional Editor Comments (optional):

The requested last adjustments were made, but the Table numbers were not adjusted after deleting Table 2. This should be done in the final version. Then the manuscript can be fully accepted.

Reviewers' comments:

All adjustments were made, but the Tables need to be numbered correctly.

---

## [Editor Report · Acceptance letter]

12 Oct 2020

PONE-D-20-24160R2 

The implementation of a rapid sample preparation method for the detection of SARS-CoV-2 in a diagnostic laboratory in South Africa 

Dear Dr. Marais:

I'm pleased to inform you that your manuscript has been deemed suitable for publication in PLOS ONE. Congratulations! Your manuscript is now with our production department. 

Kind regards, 

on behalf of

Dr. Sylvia Maria Bruisten 

Academic Editor

PLOS ONE